# An Analysis of Acculturation Status and Healthcare Coverage for the Needs of Mental Health Service Utilization among Latinos in Mississippi, Louisiana, and Alabama

**Azad R. Bhuiyan** [1,*], **Gerri A. Cannon-Smith** [2], **Sophia S. Leggett** [3], **Pamela D. McCoy** [4], **Maria Barvié** [5] and **Ashley White Jones** [6]

[1] Associate Professor, Department of Epidemiology and Biostatistics, School of Public Health, Jackson State University, Jackson, MS 39213, USA

[2] Pediatric/PH Consultant, Innovative Health Strategies, Brandon, MS 39043, USA

[3] Associate Professor, Department of Behavioral and Environmental Health, School of Public Health, Jackson State University, Jackson, MS 39213, USA

[4] Senior Prevention Specialist, Mississippi Department of Mental Health Substance Abuse Block Grant, Jackson State University, Jackson, MS 39213, USA

[5] Founder of Interlinguas, Limited Liability Company/Interpreter, Biloxi, MS 39530, USA

[6] Assistant Clinical Professor, Division of Education, Mississippi State University, Meridian, MS 39307, USA

[*] Correspondence: azad.r.bhuiyan@jsums.edu

**Abstract:** Background: The use of mental health services by Latinos is only 7.3%, despite the high prevalence of depression rates of between 27.0% and 38.0% in the United States. Research is limited concerning Latinos' acculturation status and healthcare coverage on mental healthcare service utilization in Mississippi, Louisiana, and Alabama. Therefore, the objective of this study is to examine the association of acculturation status and healthcare coverage with mental health service utilization in the Latino population. Methods: During 2011–2012, a Latino Community Health Needs Assessment was administered by a trained bilingual interviewer using participants' preferred language. Four hundred and eleven community members and leaders participated in the study. Acculturation status and self-reported mental health service utilization were retrieved from the survey instrument. Multivariate logistic regression analyses were performed. Results: In multivariate logistic regression that included gender, education level, healthcare coverage, depression, and acculturation status, individuals with a high acculturation score (3–5) were 1.53 times more likely to utilize mental health services compared to those with a low acculturation score (0–2). Individuals with healthcare coverage were 2.75 times more likely to utilize mental health services compared to those with not having healthcare coverage. Healthcare coverage is only a significant determinant of mental health service utilization. Conclusions: This result underscores the importance of having healthcare coverage for the need of mental health service utilization. Future research should consider the impact of acculturation and healthcare coverage on mental health service utilization.

**Keywords:** mental health service utilization; Latinos; acculturation status

---

## 1. Introduction

Hispanics/Latinos are the largest and fastest growing minority population in the United States (US). The United States Census Bureau projects that this population is expected to double from 53.3 million in 2012 to 128.8 million in 2060 [1]. According to the 2010 United States Census, people were considered as Hispanic or Latino who identified Latino or as one of the specific categories listed on the

census questionnaire (Mexican, Argentine, Cuban, Colombian, Puerto Rican, Dominican, Costa Rican, Guatemalan, Honduran, Nicaraguan, Panamanian, Salvadoran, Bolivian, Spanish American, Chilean, Ecuadorian, Paraguayan, Peruvian, Uruguayan, and Venezuelan). This minority group has seen steady population growth in the southern tri-states: Mississippi, from 1.4% to 2.9%; Louisiana, from 1.7% to 4.1%; and Alabama, from 2.4% to 4.5%, from 2000 to 2012 [2]. The prevalence of office-based visits for diagnosed depression among Latinos in a national sample over the 12 years examined (1992–1997 and 2003–2008) increased by 37.5% [3]. Moreover, the overall prevalence of depression symptoms in Latinos ranges from 22% among those who identify as Mexican, to 38.0% among Puerto Ricans [4]. Despite the high prevalence of depression and depression symptoms, a lower utilization of mental health services (MHS) by Hispanics or Latinos is noted among many studies [5–12]. For example, only 7.3% of Hispanic or Latino people used any MHS in 2011 in the United States, with prescription drugs use the most frequently, followed by outpatient visits [11]. Many barriers to MHS use among this population have been identified, such as nativity, socio-economic status (SES), the length of time of stay in the US, language preference, cost of health services, healthcare coverage, stigma about mental health disorders, individuals' concern about discrimination and access to medical care, and the dramatic loss of income and the acculturation process [13,14]. Acculturation status is defined as the multidimensional socialization process through which individuals from one culture adopt the attitudes, values, customs, beliefs, and behavior of another culture [15,16]. Acculturation status is reported to be associated with improved access to care and preventive services [17]. Though there is not one specific measure of acculturation (a complex phenomenon), nativity or generational status, the length of residence in the United States and language are the most predominantly used measures in public health research [15–20].

Although theory and framework influences research on acculturation, multiple layers of influence of acculturation emerged in the literature [21,22]. With regard to healthcare coverage, the Department of Health and Human Service reported that Latinos had a lower rate of healthcare coverage than that of any other race or ethnicity. In total, 19.5% of the Latino population were not covered by healthcare, as compared to 6.3% of the non-Latino white population in 2015 [23]. It also reported that health is influenced by acculturation, healthcare coverage, and access to preventive services. Research is limited and understudied concerning the factors related to MHS utilization by Latinos in Mississippi, Louisiana, and Alabama. It was hypothesized that acculturation and healthcare coverage were independently associated with MHS utilization. Therefore, this study examines the association of acculturation levels and healthcare coverage with MHS utilization, controlling for SES in this Latino population.

## 2. Methods

### 2.1. Recruitment

Four hundred and eleven (411) Latino community members and leaders ≥18 years of age participated in the Hispanics/Latinos Community Health Need Assessment survey. Among the participants, 33.6% (138) were from Mississippi (MS), 31.9% (131) were from Alabama (AL), 32.2% (132) were from Louisiana (LA) and 2.4% (10) were missing a state of residency designation. Because of the small percentage of Latinos residing in these states, convenience sampling was used to capture Latinos through faith-based, community service, and healthcare partners and organizations of various counties of the tri-states. Community participants were queried regarding their community health priorities, health status, healthcare utilization, access to care, and quality of healthcare.

1.   Inclusion criteria: Latino adults residing in this tri-state were included.
2.   Exclusion criteria: Individuals with missing information were excluded from this study.

### 2.2. Interview

Initially, in 2010, a pilot study was conducted for the Hispanics/Latinos Health Need Assessment survey, with 110 participants in Mississippi using the Pew Research Center Healthcare survey tool.

After receiving feedback from community stakeholders in the pilot study, the survey questionnaire was modified and extended to Mississippi, Alabama, and Louisiana from 2011 to 2012. A trained bilingual interviewer was employed to conduct an interview using participants' preferred language (English/Spanish). Before each interview, a short briefing about the study was discussed, and then written consent was taken from each participant. At the interview's conclusion, the participant was asked a series of demographic questions. Each interviewee received a $25 Target gift card for completing the interview. An interview ranged 25–30 min for each participant.

### 2.3. Data Collection

After the interview, data collected by the interviewers were digitally recorded in the Microsoft Excel (2013) and IBM SPSS statistics 24 (IBM Corp. Released 2016. IBM SPSS Statistics for Windows, Version 24.0. Armonk, NY: IBM Corp). Two graduate research assistants entered data into the system. Members of the study team reviewed data, validated them against a hard copy, and de-identified the data before reporting study findings.

### 2.4. Variables of Interest

The need for Mental Health Services was the outcome variable in this study. Participants were asked to answer the following question: During the past 12 months, have you seen or talked to a mental health professional (for example, a psychiatrist, psychologist, psychiatric nurse, or social worker) about your mental health? If participants answered, "yes," they were considered to have needed MHS utilization during the past 12 months, and those who replied "no" were considered not to have needed MHS utilization during the past 12 months.

### 2.5. Independent Variables

Acculturation level was the main independent variable. Language preference, duration of stay in the US and being US-born are used as proxy measures of acculturation in this study. Based on a previously published article from the Multi-ethnic Study of Atherosclerosis (MESA) [20], an acculturation score was developed from the proxy measurement of the nativity, length of stay, and language preferences. A score of 0–3 was assigned for nativity combined with years in the US (US-born = 3, foreign-born and lived in the US ≥20 years = 2, foreign-born and lived in the US 10–19 years = 1, and foreign-born and lived in the US <10 years = 0). A score of 0–2 was assigned to the language spoken at home (English = 2, English and Spanish = 1 and Spanish languages = 0). These scores were totaled to obtain the acculturation score, ranging from 0 (least acculturated) to 5 (most acculturated). We divided the acculturation score as low (0–2) and high (3–5), based on previous research and sample distribution. Immigration status was not asked, as it was a sensitive issue.

Healthcare coverage was assessed by asking participants if they had healthcare coverage all the time for the last 12 months, had a time without healthcare coverage, or never had healthcare coverage.

Depression: Depression was assessed from the following question, "had a doctor or other healthcare provider ever told you that you had depression?"

Health Status: Participants were asked to describe their health status ranging from excellent, very good, good, and fair, to poor.

### 2.6. Sociodemographic Variables

Age, gender, country of origin, and SES were assessed as part of the survey. Age was assessed as three groups: 18–24, 25–64, and ≥65, based on the funding agency's recommendation. SES was measured by the level of education and yearly income. Education level was divided into four groups: no education/elementary, high school/general education diploma vocational school/college, and university degree. Income level was assessed using three groups: <$20,000, $20,000–$39,999, and ≥ $40,000. By ethnic group, participants were divided into two groups. An individual who reported Mexico as their country of origin was considered Mexican. All others were considered other Latinos.

## 3. Data Analysis

All statistical analyses were conducted using SPSS, version 25. Descriptive analysis was conducted to characterize the study population. Spearman correlation was assessed to check the correlation matrix between the dependent variable and independent variables. Multivariate logistic regression analysis was conducted to examine the acculturation and healthcare coverage, controlling for SES. Thus, we used analysis of the total population to avoid model violation and fitness tests. Interactions were tested between acculturation and ethnicity and healthcare coverage. There was no interaction (*p*-value= 0.97) noted between ethnicity and acculturation for predicting the needed for MHS utilization. There was also no interaction between acculturation and healthcare coverage (*p*-value = 0.88) for predicting the needed for MHS utilization. For all analyses, differences were considered statistically significant if $p < 0.05$, and a 95% confidence interval (CI) was not bound by one.

## 4. Results

Table 1 shows the socio-demographic characteristics of the study participants. Out of 411 participants, 34.4% were from Mississippi (MS), 32.9% were from Louisiana (LA), and 32.7% were from Alabama (AL). Thirty-four percent were Mexican Latinos, and 64% were other Latinos (Puerto Rican 4%, Cuban 4%, Honduran 20%, Guatemalan 8%, Nicaraguan 3%, Peruvian 3%, Colombian 3%, Salvadoran 3%, other 16%). The majority of participants (57%) were females. Most of the participants (77%) were between 25 and 64 years of age. Nearly 38% had a high school/GED level of education. Approximately half of the participants (52%) reported an annual income of below $20,000. Sixty-four percent of participants never had healthcare coverage. Forty-four percent of participants had been living in the US for 20 years or more; 30% for 10-19 years, and 26% for less than ten years. Seventy-nine percent preferred the Spanish language; 15%, English; and 6%, both. Eleven percent were born in the US. Only 18% had a higher acculturation score (3-5 scores) by definition. In total, 10.4% of participants were told by a healthcare professional that they might suffer from depression, and 9.6% of people had needed MHS. The majority of participants (61.7%) considered their health to be in good condition, and most participants were married or had a partner (63.5%).

**Table 1.** Latinos' characteristics of a study population in the Tri-States.

| Variables | Number | Percent |
|:---:|:---:|:---:|
| States | —— | — |
| Mississippi | 138 | 34.4 |
| Louisiana | 132 | 32.9 |
| Alabama | 131 | 32.7 |
| Ethnic group | —— | —— |
| Mexican Latinos | 145 | 36 |
| Other Latinos | 257 | 64 |
| (Puerto Rican 4%, Cuban 4%, Honduran 20%, Guatemalan 8%, Nicaraguan 3%, Peruvian 3%, Colombian 3%, Salvadoran 3%, other 16%) | | |
| Gender | | |
| Female | — | —— |
| Male | 235 | 57 |
| Age category | 176 | 43 |
| 18–24 | — | — |
| 25–64 | 62 | 15 |
| ≥65 | 316 | 77 |

**Table 1.** *Cont*.

| Variables | Number | Percent |
|---|---|---|
| Education completed | 26 | 6 |
| No education/Elementary | —— | —— |
| High School/GED | 122 | 30 |
| Vocational School/College/University | 152 | 38 |
| Household income | 129 | 32 |
| Less than 20,000 | —— | —— |
| 20,000–39,999 | 195 | 52 |
| Above 40,000 | 118 | 31 |
| Healthcare coverage | 65 | 17 |
| Healthcare coverage for the past 12 | —— | —— |
| months | 99 | 25 |
| Had healthcare coverage | 45 | 11 |
| Never had healthcare coverage | 259 | 64 |
| Components of acculturation | —— | —— |
| Born in the US | —— | —— |
| The US born | 45 | 11 |
| Non-US born | 355 | 88 |
| Duration of stay in the US | — | —— |
| Staying 20 or more years | 102 | 44 |
| Staying 10–19 years | 118 | 30 |
| Staying less than 10 years | 117 | 26 |
| Language preference | —— | —— |
| Spanish | 309 | 79 |
| English | 57 | 15 |
| Both | 26 | 6 |
| Acculturation status | —— | —— |
| Low Acculturation (0–2 score) | 336 | 82 |
| High Acculturation (3–5 score) | 73 | 18 |
| Depression | 41 | 10.4 |
| Mental health service utilization | 39 | 9.6 |
| Health status | —— | – |
| Poor/Fair | 100 | 24.6 |
| Good/very good | 251 | 61.7 |
| Excellent | 56 | 13.8 |
| Marital status | — | — |
| Married/Partner | 257 | 63.5 |
| Divorce | 68 | 16.8 |
| Never married | 80 | 19.8 |

Table 2 provides Spearman's correlation matrix. The need for MHS utilization was significantly associated with gender: male vs. female (r = 0.14, *p*-value = 0.004), education level (r = 0.13, *p*-value = 0.009), healthcare coverage (r = 0.19, *p*-value = 0.001), acculturation level—low vs. high (r = 0.11, *p*-value = 0.03)—and depression (0.34, *p*-value 0.001). On the other hand, a high acculturation level was significantly associated with ethnicity—Latino vs. Mexican (r= −0.14, *p*-value = 0.005)—older people (r= −0.14, *p*-value = 0.004), a higher education level (r = 0.42, *p*-value = 0.001), a higher income (r = 0.32, *p*-value = 0.001), healthcare coverage (r = 0.32, *p*-value = 0.001), and marital status (r = 0.24, *p*-value = 0.001).

**Table 2.** Correlation Matrix between mental health services (MHS) and Independent Variables.

| Variables | Ethnicity Other Latinos vs. Mexican Latinos | Gender Male vs. Female | Age Category | Education | Income | Marital Status | Healthcare Coverage | Acculturation Low vs. High | Depression | Health Status |
|---|---|---|---|---|---|---|---|---|---|---|
| MHS *p*-value | −0.85 0.09 | 0.14 0.004 | 0.06 0.21 | 0.13 0.009 | 0.02 0.67 | 0.08 0.09 | 0.19 0.001 | 0.11 0.03 | 0.34 0.001 | −0.78 0.12 |
| Acculturation Low vs. High *p*-value | −0.14 0.005 | 0.09 0.06 | −0.14 0.005 | 0.42 0.000 | 0.32 0.001 | 0.24 0.001 | 0.32 0.000 | 1 0.001 | −0.08 0.107 | 0.07 0.15 |

Table 3 depicts the predictors of MHS utilization for those of Latino ethnicity. The multivariate model included gender, education level, healthcare coverage, depression, and acculturation levels. In this analysis, individuals with higher levels of acculturation (vs. low) were 1.53 times (95% CI: 0.58–4.00) more likely to utilize MHS. Individuals with healthcare coverage (vs. no coverage) were 2.75 times (95% CI: 1.23–6.19) more likely to utilize MHS. Individuals with self-reported depression were 10.58 times (95% CI 4.50–24.39) more likely to utilize MHS. The model explained 24.9% of the variability of the MHS utilization of these variables. The model was also a good fit, with a Hosmer–Lemeshow goodness of fit test value of 3.8 and a *p*-value of 0.80. These odds ratios were adjusted for gender and education. Of note, we also ran the analysis by using acculturation as a continuous variable. It did not show a significant variable as CI bound one (AOR: 1.02, 95% (0.78–1.33)). We could not establish the mediator role of acculturation, as only 10% of responders utilized MHS. The Hosmer–Lemeshow goodness of fit test for the logistic model was not a good fit, as some cells were zero (data not shown).

**Table 3.** Independent Factor of Mental Health Service Utilization by Latinos Population in the Tri-States.

| Variables* | β Coefficient | Standard Error | Wald Chi-Square | *p*-Value | Odds Ratio | 95% CI |
|---|---|---|---|---|---|---|
| Acculturation | — | — | — | — | — | — |
| Low | Ref | Ref | Ref | Ref | Ref | Ref |
| High | 0.42 | 0.49 | 0.74 | 0.39 | 1.53 | 0.58–4.00 |
| Healthcare coverage | 1.01 | 0.41 | 6.01 | 0.01 | 2.75 | 1.23–6.19 |
| Depression | 2.36 | 0.41 | 29.74 | 0.001 | 10.48 | 4.50–24.39 |

*Variables in the equation: Acculturation level, healthcare coverage, education, gender, and depression. Significant variables from Table 2 of the correlation matrix were entered in the model. Model fit test: Hosmer–Lemeshow goodness of fit Chi-square test = 3.79, df = 7 and *p*-value = 0.80. Model summary: -2 Log likelihood 191.55, Nagelkerke R Square 24.9%.

## 5. Discussion

In the tri-state area of MS, LA, and AL, our findings suggest that acculturation status is not significantly associated with MHS utilization, after controlling for SES and healthcare coverage. Although previous studies found acculturation is associated with a higher utilization of mental health service [8] at the national level, literature is lacking in MHS service utilization by Latinos in this region. A previous study used individual components of acculturation—for example, ethnicity, language preference, and duration of stay in the United States. We used a multi-dimensional model of acculturation using three measures: language preference, birth status, and length of time in the United States. High acculturation status is associated with healthcare coverage, which is noted in the correlation analysis. This association is consistent with literature findings [17].

Healthcare coverage is the strongest predictor for needing MHS utilization among Latinos living in the tri-state area, after controlling for gender, education, and depression status and acculturation level. The model accounted for 24.9% of the variability of MHS utilization. The result shows that healthcare coverage increases the odds of MHS utilization by 2.75 times among Latinos. This finding is consistent with the previous studies that assessed MHS utilization by Latinos, as well as our previous study predicting dental care service utilization [8,24]. In our previous study, healthcare coverage was the strongest predictor of dental health service utilization in the tri-states. According to the US Census Bureau, in 2012, 33.4% (15.8 million) of Latinos were without healthcare at the national level, and 18.2-19.7%of Latinos had no healthcare coverage in the southern region in 2009 [25]. In this study, we found that 64% of Latinos were without healthcare coverage. This finding is self-reported from a nonrandom sample population and may lead to response bias and selection bias. The 2013 Report on Healthcare Coverage by Race and Ethnicity confirmed that over half of those "without healthcare coverage" are people of color, most of whom are adults [26].

The 2007 Pew Healthcare Survey revealed that healthcare coverage was an important barrier to Hispanic/Latino's access to a healthcare system [27]. Healthy People 2020 target for increasing the proportion of persons with healthcare coverage is 100% [28]. The 2010 Affordable Care Act (ACA)

facilitates and increases healthcare coverage through Medicaid expansion and the creation of private healthcare coverage with a tax credit [29–32]. The current administration seeks to undermine and repeal the ACA. Therefore, ACA needs to be reconsidered for increasing healthcare coverage, and reducing healthcare disparities in mental health in this community.

Our study showed that acculturation varied by state, and there was a correlation between acculturation and healthcare coverage. Our study showed that high acculturation was noted in LA (54.9% compared to 28.2% in MS and 16.9% in AL). In 2017, the Louisiana Department of Health announced that more than 400,000 individuals had enrolled in coverage under the state's Medicaid expansion [33]. Mississippi and Alabama have yet to consider legislation to expand Medicaid for residents. In Mississippi alone, this would increase coverage to approximately 300,000 residents who currently fall within the gap. Expansion in these states would greatly affect the accessibility of healthcare coverage and potentially impact the mental health services utilization of Latinos residing in these states.

There are several limitations to this study. First is the inability to differentiate Latinos by subgroups. Multiple countries were identified as the country of origin. Because of the small number of participants from those countries, they could not be divided into Latino subgroups. Second is the the bias introduced by self-reporting. It could be improved by an objective measure of depression. Studies also suggest that the utilization of MHS varied by Latino subgroups. Latinos of Puerto Rican origin were found to be more likely to suffer from mental disorders and were more likely to use MHS than other Latino subgroups [6,12]. Secondly, the study design used a retrospective measure of MHS use. This might have created recall bias in the findings because of recall reporting. The validity measures of self-reported MHS utilization were not available. Third, this is a convenience sample, and cross-sectional analysis cannot determine causal inferences. As mentioned, stigma may play a role in MHS utilization, but it is was not analyzed in this study.

A major strength of this study is that it is the first study to our knowledge that has assessed needs for MHS utilization among Latinos in the southern United States, specifically in Mississippi, Louisiana, and Alabama. This study examines the MHS utilization by acculturation level and healthcare coverage controlling for SES. Thus, it makes an important contribution to the existing literature in the southern US

## 6. Conclusions

Healthcare coverage is the strongest predictor for the need for MHS utilization supported by previous studies. The association of acculturation and healthcare coverage with MHS utilization among Latinos in the tri-states warrants further robust investigation.

## 7. What Are the Implications for Public Health Research?

This study examined the independent factor associated with mental health service utilization. It underscores the importance of having healthcare coverage. Healthcare coverage is necessary for Latinos to access mental health services irrespective of their acculturation level and SES status. Future research should consider the impact that acculturation and healthcare coverage have on MHS and investigate ways to reduce barriers and facilitate access to MHS.

**Author Contributions:** A.R.B. designed, analyzed, and prepared the manuscript. G.A.C.-S. and S.S.L. are the Principle investigators of the funded project. P.D.M. was the study coordinator. They provided substantial revisions to the article. M.B. was responsible for data collection, and A.W.J. was a graduate student who aided in data collection and article revisions.

**Funding:** Supported in part by a grant with the University of Mississippi Medical Center's Institute for Improvement of Minority Health and Health Disparities in the Delta Region and funded by the Department of Health and Human Services' Office of Minority Health (Prime Award Number 1 CPIMP091054-01-00). The Delta Regional Institute's charge is to eliminate health disparities. The initial part of this study was presented at The National Institute on Minority Health and Health Disparities Grantees' Conference, formerly known as the International Symposium on Minority Health and Health Disparities, Maryland, 1–3 December 2014.

**Acknowledgments:** This study was a joint effort of Hispanic and Latino community stakeholders, leaders, and individuals whose co-operation is gratefully acknowledged. We are especially grateful to the bilingual interviewer, Maria Georgie Barvié, and the Latino study participants in Mississippi, Alabama, and Louisiana. We thank our collaborator and research team members.

**Conflicts of Interest:** All surveys were conducted after receiving approval from the Jackson State University Institutional Review Board.

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
