# Peer review of "An Analysis of Acculturation Status and Healthcare Coverage for the Needs of Mental Health Service Utilization among Latinos in Mississippi, Louisiana, and Alabama"

_psych, doi:10.3390/psych1010035_

Round 1

Reviewer 1 Report

This study is investigating the mental health service use of Latinos in MS, LA, and AL. While an important topic, this research fails to meet the scientific rigor and thinking that would be necessary. 

Introduction: The introduction does not give a satisfactory background in regards to the mental health incidence rate and service needs of Latino populations. It also does not provide a satisfactory rationale for acculturation, healthcare coverage, and mental health service use are related. There seems to be no theoretical underpinnings to the study, nor a hypothesis, resulting in a rather vague fishing expedition. 

Methods: The method section is reflective of the lack of direction shown in the introduction. We don't know if there was any quality check between interviewers, nor is there any explanation why there is only one depression question, and the question about unmet mental health service use is not described (only alluded to in other sections). 

There is also no explanation why Mexicans are in their own category. 

Results: The demographics are jumbled, in no logical order, and contain non-demographic variables. 
Most importantly, it is unclear why the authors did not conduct the analysis on those that reported having depression. Again, a hypothesis would have driven the analysis in a more logical fashion. 

Discussion: Not surprisingly, the paper concludes with a weak discussion section with no hints towards any hypothesis and theorized relationship between variables. 

Author Response

Thank you so much for carefully reviewing our manuscript. We really appreciate it. We responded to your comment and updated manuscript based on your comment and suggestion

Reviewer 2 Report

Please see comments and suggestion as included in the pdf.

I did not receive any tables or figures and therefore, could not review the results fully.

The survey questionnaire should be added in appendix,

Line 47; enter "reported to be" after 'is'

Line 68: Should 'Variables' be replaced by "Individuals" as the heading is Recruitment / inclusion / exclusion?

Line 128: Replace 'was seeking'  by "utilized" and delete 'utilization'

Line 139: As only Latinos were included per the inclusion criteria, how comes, that only 64% are Latinos? If this relates to the differentiation of Mexican latinos versus other Latinos, it should be described more clearly.

Line 145/146: (1) What does it mean "10.45 were depressed"? Does it mean, they were actually depressed, or they were told by a healthcare professional they may suffer from depression?
(2) If 10.4% had depression and 10% utilized MHS, that means that most of the depressed patients were treated?

Line 156-158: sentence and meaning unclear (perhaps leave away if not relevant to the paper)

Line 186: Delete 'it also showed that'

Lines 189, 193, 195, 196, 198: Correct references

Lines 201-201: What exactly is the logic here? What's 'the alternative plan' in this context?

Line 214: I would suggest to formulate this more clearly as shortcoming (self-report) and a proposal how to improve this in subsequent studies

Line 221: Replace 'it is the out of the scope of' by "was not analyzed in"

Line 233: Replace 'provide' by "be able to access"

Line 233 /234: define this better. what do you mean with 'in this community'

Author Response

Dear reviewer;

We really appreciate your time to spend on this manuscript. We revised it based on your comment and suggestion. Tables were missing during upload I believed. I will check carefully this time during uploading. Thank you so much for your feedback. It helps a lot to improve the manuscript. 

Take care. 
